# COVID-NURSE: evaluation of a fundamental nursing care protocol compared with care as usual on experience of care for noninvasively ventilated patients in hospital with the SARS-CoV-2 virus—protocol for a cluster randomised controlled trial

David A Richards [1,2] Holly VR Sugg,[1] Emma Cockcroft,[1] Joanne Cooper,[3] Susanne Cruickshank,[4] Faye Doris,[1] Claire Hulme,[1] Phillipa Logan,[5] Heather Iles-Smith,[6] G.J Melendez-Torres,[1] Anne Marie Rafferty,[7] Nigel Reed,[1] Anne-Marie Russell [8] Maggie Shepherd [9] Sally J Singh,[10] Jo Thompson Coon [1] Susannah Tooze,[8] Stephen Wootton,[11] Rebecca Abbott [1] Alison Bethel,[1] Siobhan Creanor,[12] Lynne Quinn,[12] Harry Tripp,[12] Fiona C Warren,[1] Rebecca Whear,[1] Jessica Bollen,[12] Harriet A Hunt [1] Merryn Kent,[1] Leila Morgan,[8] Naomi Morley,[1] Lidia Romanczuk[9]

For numbered affiliations see end of article.

**Correspondence to**
Professor David A Richards;
D.A.Richards@exeter.ac.uk

## ABSTRACT

**Introduction** Patient experience of nursing care is correlated with safety, clinical effectiveness, care quality, treatment outcomes and service use. Effective nursing care includes actions to develop nurse–patient relationships and deliver physical and psychosocial care to patients. The high risk of transmission of the SARS-CoV-2 virus compromises nursing care. No evidence-based nursing guidelines exist for patients infected with SARS-CoV-2, leading to potential variations in patient experience, outcomes, quality and costs.

**Methods and analysis** we aim to recruit 840 in-patient participants treated for infection with the SARS-CoV-2 virus from 14 UK hospitals, to a cluster randomised controlled trial, with embedded process and economic evaluations, of care as usual and a fundamental nursing care protocol addressing specific areas of physical, relational and psychosocial nursing care where potential variation may occur, compared with care as usual. Our coprimary outcomes are patient-reported experience (Quality from the Patients' Perspective; Relational Aspects of Care Questionnaire); secondary outcomes include care quality (pressure injuries, falls, medication errors); functional ability (Barthell Index); treatment outcomes (WHO Clinical Progression Scale); depression Patient Health Questionnaire-2 (PHQ-2), anxiety General Anxiety Disorder-2 (GAD-2), health utility (EQ5D) and nurse-reported outcomes (Measure of Moral Distress for Health Care Professionals). For our primary analysis, we will use a standard generalised linear mixed-effect model adjusting for ethnicity of the patient sample and research intensity at cluster level. We will also undertake a planned

### Strengths and limitations of this study

► This study is the first randomised controlled trial of a fundamental care clinical nursing protocol for patients infected with the SARS-CoV-2 virus.

► The intervention relates specifically to patients with the SARS-CoV-2 virus admitted to inpatient wards who are not invasively ventilated.

► The intervention programme theory will enable generalisation of our findings to other environments such as care homes, patients with other conditions requiring isolation, and global health systems.

► The trial coprimary outcomes require patient participants to have capacity to consent and report their experience of care, and, therefore, we will be unable to collect and report data from patients who lack this capacity.

► In common with other cluster randomised controlled trials of behaviour change interventions, we are unable to blind nurses, participants or data collectors to trial arm allocation

subgroup analysis to compare the impact of patient-level ethnicity on our primary and secondary outcomes and will undertake process and economic evaluations.

**Ethics and dissemination** Research governance and ethical approvals are from the UK National Health Service Health Research Authority Research Ethics Service. Dissemination will be open access through peer-reviewed scientific journals, study website, press and online

media, including free online training materials on the Open University's FutureLearn web platform.

**Trial registration number**  ISRCTN13177364; Pre-results.

## BACKGROUND

Patient experience of nursing care is correlated with safety, clinical effectiveness, care quality, treatment outcomes including mortality and overall service use.[1–7] Nursing care is a key determinant of patient experience[8 9] and satisfaction.[10] Effective nursing care includes the establishment of compassionate nurse–patient relationships known as 'relational nursing care' in order to facilitate the physical and psychosocial care work undertaken by nurses to meet the fundamental care needs of patients such as nutrition, hydration, skin integrity, personal care, mobility, hygiene, breathing, elimination and mental well-being.[11 12]

Measuring patient experience is analogous to collecting a patient-reported outcome measure (PROM)[13] in that both are '*a measurement based on a report that comes directly from the patient (ie, study subject)*'.[14] However, whereas a PROM captures '*the status of a patient's health condition without amendment or interpretation of the patient's response by a clinician or anyone else*',[14] a patient-reported *experience* measure (PREM) is a measure of a patient's perception of their personal experience of the healthcare they have received.[15] Two PREMs for evaluating patients' experiences of nursing have been developed and tested; the Relational Aspects of Care Questionnaire—RACQ)[8 16] and the Quality from the patient's perspective—QPP.[17 18]

The combination of COVID-19 symptoms and high risk of transmission of the SARS-CoV-2 virus poses unique challenges for physical, psychosocial and relational nursing care. Previously, the 2003 SARS outbreak demonstrated that nurses' ability to provide compassionate fundamental care was compromised: '*The establishment/maintenance of therapeutic nurse-client relationship required additional time given the barriers of mask, gloves and gowns.* p4[19] '*Restrictions on visitors were difficult for staff because family members are usually involved in the social, psychological and, to some extent, physical care of patients*'. p6[20] '*Interaction time decreased, and patients began to feel more abandoned*'. p28[21]

During the development of our trial protocol described below, we found no international evidence-based guidelines for nursing hospitalised patients with the SARS-CoV-2 virus, the majority of whom are not invasively mechanically ventilated. This leads to the potential for variations in patient experience, treatment outcomes, care quality and costs, as reported by nurses who responded to our survey, also described below. Although individual nurses, teams and organisations reported rapidly adapting their procedures to nurse these patients, compared with usual nursing care, we as yet do not know how to optimise nursing care for the specific challenges of nursing patients with COVID-19, including mitigating the impact of wearing personal protective equipment (PPE), nor

how effective these adaptations are in terms of these important outcomes.

Between July and October 2020, we developed a clinical nursing protocol. We undertook a rapid review[22] of published literature to identify the evidence for the effectiveness of, and barriers to, fundamental nursing care procedures in patients with the SARS-CoV-2 virus or other conditions requiring isolation, in terms of overall patient experience, care quality, functional ability and treatment outcomes—PROSPERO registration number: CRD4202020091.[23] We also ran a UK-wide survey of nurses and nonregistered care staff working with hospitalised patients with the SARS-CoV-2 virus who were not invasively ventilated to identify respondents' views on 'missed care',[24 25] barriers to care and innovation strategies respondents had adapted to meet the fundamental care needs of such patient. We obtained research governance approval from the Health Research Authority (IRAS project ID: 287288, protocol number 1920/31) and ethical approval from the University of Exeter research ethics committee (no: 20/07/256) for this survey. We then convened three consensus development panels[26–28] of nurses, and patients with experience of hospitalisation with the SARS-CoV-2 virus, to agree the content and design of the clinical nursing protocol. We will report the results of our systematic review and survey elsewhere.

In the COVID-NURSE trial, our objectives are: (a) to compare care as usual and an evidence-based nursing protocol for patients with the SARS-CoV-2 virus against care as usual, in terms of patients' reported experience of transactional and relational nursing care, care quality, treatment outcomes and costs and (b) to compare the effects of using the clinical protocol on nurses' moral distress

## METHODS: PARTICIPANTS, INTERVENTIONS AND OUTCOMES
### Design

We will undertake a cluster randomised controlled superiority trial[29 30] with embedded iterative process[31] and economic evaluations[32] between 02 November 2020 and 18 October 2021). Our initial funded, approved and registered design was a rapid-cycle cluster randomised controlled trial[33] with three review 'waves'. Like adaptive trial designs, rapid cycle trials embed *a priori* opportunities into trial designs to enable review and minor adaptions to interventions between waves. We also planned to pair match 18 trial sites according to two variables with a potential impact on outcomes: (1) ethnicity, where being of black, Asian and minority ethnicity (BAME) increases the risk of SARS-COV-2 infection compared with white individuals and where people of Asian ethnicity may be at higher risk of intensive care unit (ICU) admission and death[34] and (2) research intensity which is associated with improved hospital outcomes.[35] We planned to classify ethnicity according to the latest census data for the local authority in which a hospital is located (low (<68%) vs medium (68% to 74%) vs high (>74%) non-BAME

categories based around an overall mean of 71% non-BAME admissions. We planned to classify research intensity according to two categories: research intensive versus not research intensive, where research intensive is defined as ≥0.1 unique recruiting studies in the 2019–2020 Q4 open data from National Institute for Health Research (NIHR) divided by number of inpatient elective admissions in January–March 2020, and nonresearch intensive is <0.1.

Since submitting this protocol for peer review by this journal, experience enrolling wave 1 sites during October–January 2021 taught us that our ideal situation is subjected to considerable stress as sites adapt to a rapidly changing situation on the ground, facing significant clinical and research pressures. Although we successfully recruited three pairs (six sites) of sites matched as above for our planned first wave during October–December 2020, operational difficulties caused by (a) the trial not being prioritised for National Health Service (NHS) research infrastructure resources and (b) the impact of the second wave of the SARS-COV-2 virus on nursing sickness levels, led to very significant delays to site opening. Our six wave one sites were unable to start data collection concurrently, delaying our planned wave 1 review. Furthermore, we recruited several other sites for waves 2 and 3 for which we could not find a match on the variables listed above. These issues threatened to prevent us from recruiting to time and target within the funded period (July 2020–April 2021). Therefore, in order for the trial to recruit sufficient sites and participants and for us to retain our ability to review and make minor amendments to the intervention in response to process data on fidelity and acceptability, we amended the design to a simple cluster randomised trial, with an embedded iterative process evaluation. As a consequence, as part of the statistical analysis plan, we will also adjust for ethnicity of the patient sample at cluster level and, where patient outcomes are tested, at patient level rather than prerandomisation. Finally, we revised our sample size calculations, see below and obtained additional funding to continue the trial for a further 6 months from the previous planned end date of 18 April 2021. We began data collection on 18 January 2021 and plan to continue until the end of August 2021.

## Setting
We will recruit 14 UK district-general and teaching hospital NHS Trusts (clusters). Participants will be recruited from one or more wards at each site where patients are being treated for infection with the SARS-COV-2 virus.

## Participant eligibility criteria
Patients who are not invasively ventilated, aged ≥18 years, currently hospitalised and being treated for infection with the SARS-CoV-2 virus or recently discharged after such treatment and who have received nursing care for a minimum of 72 hours during their admission; registered nurses and nursing care workers working under the supervision of registered nurses caring for patients hospitalised

and treated for infection with the SARS-CoV-2 virus. Participants must be able to give informed consent (online supplemental appendices 1 and 2). We will provide translation facilities for participants unable to understand and speak English. We will recruit participants specifically from wards allocated to the care of patients admitted for treatment of COVID-19 symptoms.

## Interventions
### Experimental intervention
Care as usual and the clinical protocol developed as described above.

In order to preserve intervention blinding across trial control sites, we have not included the full clinical guideline here. To do so would potentially unblind nurses working in control cluster sites. Although it is an ethical principle to make a full intervention protocol available for peer and reader scrutiny, there is a competing and equal ethical principle to preserve control cluster integrity and prevent contamination of control sites. This ensures that all participants in a trial contribute their data to a study that is neither compromised nor subjected to significant potential biases. Trials with compromised blinding are often rejected in systematic reviews or lead to uncertain conclusions. Below is a summary, therefore. However, we commit to making all written and online materials available to the scientific and clinical community free of charge at the conclusion of the trial, to include a free to access Massive Online Open Course supporting the intervention. In the interim, we will refuse no reasonable request to view our guideline, having assured ourselves that a request will not compromise intervention blinding.

The protocol consists of four elements adapted from the methods used successfully in a previous cluster randomised controlled trial to significantly increase handwashing by nurses and care staff[36]: a guideline, trigger reminder posters, staff education programme and leadership from ward managers and senior nurses.

The clinical guideline consists of 26 potential strategies that can be used by nurses to address barriers to physical, relational and psychosocial nursing care identified in our survey and systematic review. In our survey,[37] eight barriers were ranked within the top five in at least one component of the three physical, relational and psychosocial care areas. These were wearing PPE, the severity of patients' conditions, inability to take items in and out of isolation rooms without donning and doffing PPE, lack of time to spend with patients, lack of presence from specialised services, for example, physiotherapists, lack of knowledge about COVID-19, insufficient stock and reluctance to spend time with patients for fear of catching the SARS-CoV-2 virus.

From this data, we grouped the guideline strategies thematically into actions which address: (a) communication with patients, with patients' significant others, between patients and their significant others, between nurses and between nurses and other members of the care team, (b) the organisation of fundamental nursing

care activities, (c) addressing the values of patients and their significant others, (d) delivering specific fundamental nursing care interventions, (e) identifying and responding to the mental health and well-being needs of patients' and their significant others.

The trigger posters contain key information on these same strategies to remind nurses to use them. The online educational resource includes video testimony from patients, carers, nurses and scientists about these same care activities and is hosted on the Open University's 'FutureLearn' platform.[38] Finally, the guideline includes advice to managers on organising care teams, educating and supporting members of their nursing teams, information on this also delivered via the 'FutureLearn' platform.[38]

We will monitor clinical protocol fidelity and acceptability as part of our process evaluation and modify our guideline, educational and leadership strategies iteratively, making minor adjustments to these elements according to findings from our process evaluation quantitative and qualitative data. We will embed any changes to the protocol in the subsequent experimental intervention sites.

### Control

Care as usual only. We will record staffing staff skill mix details alongside a description of clinical nursing procedures and organisation in place in the usual care clusters.

### Outcomes

We will collect participant-level outcome data. Our coprimary outcomes will be patient reported experience of transactional nursing care using the QPP[17 18] and relational nursing care using the RACQ.[8 16] Our secondary outcomes will include: measures of quality of care (pressure injuries, falls, medication errors) collected by the UK National Health Service Ward to Board dashboard[39]; functional ability by the Barthel Index[40 41]; treatment outcomes by the WHO Clinical Progression Scale[42]; depression by the PHQ-2,[43] anxiety by the GAD-2,[44] health utility by the EQ5D[45] and nurse outcomes by the Measure of Moral Distress for Health Care Professionals.[46 47]

### Participant timeline

We will undertake the trial during 50 weeks of intervention and rolling data collection, with data collected from patient participants admitted for treatment of SARS-CoV-2 infection and staff caring for them.

### Target sample size

Using published formulae by Hayes and Moulton,[16] we calculated our patient participant sample size based on an estimate of the minimum clinically important difference for the QPP of 0.2 and the typical within-unit standard deviation (0.6), supplied by the measure developers and an intraclass correlation coefficient (ICC) of 0.02. We have examined a number of potential scenarios, taking into account different cluster sizes and different between-cluster SD values informed by between-country

differences on measures, estimated to provide 90% and 80% power. Based on conversations about feasibility with NHS clinical leaders, with a cluster size of 60 the trial would generate over 80% power with six clusters per arm and 90% power with seven clusters in each arm. However, because power calculations in cluster trials with a small number of clusters can be especially sensitive to the approximation used for df, we used a range of alternative approaches (no small sample adjustment, Satterthwaite approximation and Kenward-Rogers approximation; cluster-level linear regression) to explore the robustness of the sample size calculations. While these tended to estimate reduced power at each number of clusters compared with a formula-based power calculation, an estimate of 14 clusters (7 per arm) generated power of greater than 80% in every case. This number of clusters is fairly robust to number of patients per cluster. Even assuming under-recruitment leading to 50 patients per site, power is maintained at greater than 80%. Therefore, we will aim to recruit 840 patient participants. Although we have not powered the sample size on nurse participant outcomes, in consultation with clinical colleagues in our investigator and advisory groups, we anticipate that around 50 registered nurses and healthcare support workers will be involved in the direct care of these patient participants leading to a recruitment target of 700 registered nurse and nursing care worker participants, including student nurses.

### Recruitment

We will recruit cluster sites via the NHS chief nurse and clinical research networks. We will monitor available data from Public Health England on case and hospitalisation rates by NHS Trust in order to identify recruitment sites where there are sufficient potential participants. We will use these data to prepare sites for involvement. Clinical research nurses will recruit patient participants from admitted patients according to the eligibility criteria above.

### Allocation

We will randomly allocate sites to the intervention and control groups by an unblinded statistician who has no role in site recruitment or data analysis to ensure allocation concealment; only the unblinded statistician will have access to the allocation list. Randomisation will use a static blocked list through the use of an externally administered, password-protected randomisation website independently developed and maintained by the United Kingdom Clinical Research Collaboration (UKCRC)-registered University of Exeter Clinical Trials Unit (ExeCTU).

### Data collection

We will collect outcome data from patient participants after they have received nursing care for a minimum of 72 hours, either during their admission or a maximum of 2 weeks postdischarge. All outcome measures will be

applied to all participants equally and collected by clinical research nurses and other good clinical practice[48] trained research staff, who will also obtain consent. It will not be possible to blind research nurses, clinicians or patient participants to allocation, although control group cluster sites will be blind to the intervention as described earlier. Statistical analysts will be blind to group allocation when receiving data to analyse. We will collect outcome data either face to face, by video/audio technology or by post following discharge according to the infection control procedures in place and participant preference. We will collect outcome data from nurses either face to face in their hospital or over the phone, according to their preference. Site level *Ward to Board* data will be collected from routine hospital statistics.

## Data management

Participants will be identified by a unique study ID. Personal identifiable data, including date of birth and NHS number, may be collected but will be stored separately to research data and will be destroyed as per applicable regulations when the project is concluded. Data will be managed by ExeCTU following General Data Protection Regulation (GDPR) and data protection guidelines and all relevant Clinical Trial Regulations. All data will be anonymised prior to publication. Data will be collected and stored electronically in accordance with the Data Protection Act 2018 and ICH GCP E6 R2. ExeCTU will use Redcap Cloud Electronic Data Capture System to collect all case report data. This system is validated to ISO27001 standards, backed up and maintained in Europe. This system is fully compliant to GDPR regulations and managed by ExeCTU. Any additional study data will be stored and backed up on the secure ExeCTU servers and maintained by ExeCTU. Data will be cleaned and validated appropriately, and a full Data Protection Impact Assessment will be undertaken along with the development of a comprehensive Data Management Plan before the first participant is recruited.

Where data are disseminated (eg, via report, presentation or publication), they will be anonymised. We will align all confidentiality and data handling with the Caldicott principles.[49] Anonymised data will be stored indefinitely on a research data storage system provided by the University of Exeter called Open access Research Exeter for archiving (http://www.exeter.ac.uk/research/open-research/policies/ore/).

## Statistical data analysis

We will analyse numerical data in Stata V.16.[50] Our primary analysis for patient and nurse outcomes will use a standard generalised linear mixed-effect model with appropriate small-sample adjustment to account for the small number of clusters, and adjusting for ethnicity of the patient sample at cluster level and, where patient outcomes are tested, at patient level as well. Sensitivity analyses for patient and nurse outcomes will use variance-weighted cluster-level summaries with adjustment for

ethnicity based on 2011 census and hospital postcode and for research intensity. Analysis of hospital outcomes will use rate ratios. Second, and focusing on patient-level outcomes, we will estimate intervention effectiveness in an exploratory analysis. This analysis will use generalised linear mixed-effects models to model time trends. Analysis will include time from implementation, reflecting growing adaptation and expertise, as well as calendar time, reflecting the course of the COVID-19 epidemic. We will also undertake a planned subgroup analysis to compare the impact of patient-level ethnicity on our primary and secondary outcomes.

## Missing data

Data are collected from patients and nurses at one time point. As a result, missingness is likely to be at item-level within scales and at scale level. Where >50% of items in a scale are completed (including 'not applicable' or 'don't know' responses), a scale score will be generated by either taking the average of remaining items (all QPP subscales, RAC-Q-14, MMD-HP) or by rescaling sum scores to the full range (Barthel Index). This is an appropriate strategy when factor loadings are homogeneous and reliability for scales is good.[51] Observations from single-item or two-item scales (WHO Clinical Progression Scale, PHQ-2, GAD-2) will be dropped from the analysis.

## Process evaluation

We will evaluate the impact of nurses' and nursing care workers' fidelity to the clinical protocol, intervention mechanisms and context of care delivery on outcomes using: a bespoke instrument for capturing fidelity to reflect the protocol components; the Culture of Care Barometer Questionnaire[52]; the NoMAD instrument for measuring implementation readiness and acceptability[53 54]; contextual data on patient acuity, staffing ratios, nurses/care staff education levels, seniority and year of qualification and other contextual variables. We will conduct interim analyses of process evaluation data on fidelity to the protocol and protocol utilisation to evaluate intervention component fidelity and implementation. We will subsequently amend the clinical protocol based on this data before embedding the new protocol in intervention sites.

We will conduct semistructured qualitative interviews with a purposive sample of nurses and care staff, using a topic guide derived from the content of the clinical protocol and our analysis of quantitative data, to explore their views on the mechanisms, impact and acceptability of the clinical protocol. This method will enable us to investigate the meaning of participants' responses, both exploring views on our predefined topics of interest and eliciting more detail on any emerging themes.[55] Interviews will be conducted using telephone or online audio-conference methods, unless face to face interviews are risk assessed as being safe and are preferred by participants. With participants' consent (online supplemental appendices 1 and 2), we will audio-record interviews and

transcribe interviews verbatim. We will use NVivo10[56] to organise the data and analyse data using Framework analysis to allow for the combination of inductive and deductive approaches in our development of themes.[57] We will use joint displays[58] to integrate the qualitative data with the quantitative outcome data in a mixed-methods analysis[59] to explore and explain variation in the outcome data. We will use this data to refine an intervention programme theory[60 61] incorporating context, mechanisms and fidelity.[31]

### Economic evaluation

We will conduct within trial cost-effectiveness analysis by adopting the NHS perspective with the main outcome measure the quality-adjusted life year (QALY) derived from the EQ-5D-5L.[32 45] We will apply the NICE threshold of £20 000 per QALY. Cost data will include staff training, length of stay, including ICU costs and COVID-19-specific interventions from baseline to discharge using a bespoke hospital care use inventory. We will apply unit costs derived from national sources including Personal Social Services Research Unit[62] and NHS Reference Costs.[63] Nonparametric bootstrapping will be used to quantify uncertainty. We will present outputs as ICERs, cost-effectiveness acceptability curves and expected net benefit.

### Monitoring

Day to day management of the trial and study will be by the chief-investigator, other coinvestigators and a research manager. We will set up an independent combined Trial Steering and Data Monitoring Committee (TSC/DMC)[48] and a patient and public involvement group.[64] We will undertake a risk assessment of the trial and produce a monitoring plan that will be commensurate with the risk identified to particularly ensure that we have input from people of colour, including from Black, Asian and other minority ethnic groups, who are at greater risk from infection with the SARS-CoV-2 virus.

### Safety reports of serious adverse events

Given this trial is not a clinical trial of an investigational medicinal product (CTIMP), the recording and reporting of non-serious adverse events (SAEs) is not required,[65] since, although likely to be common in the target population, they are unlikely to be related to either the intervention or trial participation. Likewise, by nature of the patient participants' clinical condition, we will expect some SAEs, specifically death or prolongation of hospital stay,[65] but that these will be both expected and unlikely to be related to intervention or trial procedures. As all non-CTIMP research is required to adhere to the principles of good clinical practice,[48 66] it was nonetheless agreed with the trial sponsor, research ethics committee (Health Research Authority, North East—Newcastle & North Tyneside 2 Research Ethics Committee, reference: 20/NE/0253) and independent TSC/DMC to record and review each SAE by the CI on a case-by-case basis to determine if it is related to the trial intervention or procedures. This process is overseen by the TSC/DMC.

### Patient and public involvement

The proposed study has been informed by ongoing collaborations between patients, clinicians and researchers, including in a previous NIHR-funded project ESSEntial Nursing CarE (ESSENCE) on fundamental nursing care.[67] A version of the lay summary was reviewed by six members of the 'Peninsula Patient Involvement Group', giving feedback on the broad idea of the research as well as specific points on the clarity of the summary. A patient coapplicant has been involved with the development of the proposal. We will ensure that there is a patient voice in all discussions and decisions and will ensure the integration of broader PPI activity in these decisions. We have worked with a wider group of patients with experience of hospitalisation for COVID-19 symptoms to inform the coproduction of the nursing protocol, development of all patient facing materials, finalising the nursing protocol for rapid-cycle testing, adaptions to the nursing protocol after each cycle of testing, dissemination of findings, including the production of an online training module for nurses on the FutureLearn platform. The involvement of patients will be informed by what is possible within the ongoing pandemic and to suit the situation of interested patients. Involvement will be flexible and responsive to their needs. We will adopt high standards of equality, diversity and inclusivity[68] throughout.

### Ethical issues

We will conduct this trial in such a way as to protect the human rights and dignity of the participants, as reflected in the Helsinki Declaration.[69] We have obtained governance and ethical approval from the Health Research Authority National Research Ethics Service (IRAS project ID 288479; REC reference: 20/NE/0253). Participants will not receive any financial inducement to participate. We will invite eligible patients and nurses to participate in data collection and will only collect data from participants who individually consent. Participants may withdraw from the study at any time without prejudicing their treatment, care or employment. We will conform to Good Clinical Practice Guidelines, data protection and freedom of information acts. We prospectively registered the trial with the ISRCTN registry on 9 September 2020, before our intended data collection start date. We have registered protocol amendments when required.

### Outputs and dissemination

We will disseminate our results through publication in peer-reviewed scientific journals, our study website and other press and online media. Manuscript authors will be those considered to have made a substantive intellectual contribution to the study. The investigators and relevant authorities will have access to the trial dataset. Furthermore, we will store anonymised research data and outputs in the University of Exeter's Open Research

Exeter repository (https://ore.exeter.ac.uk/repository/) in order to facilitate open access to, and the impact of, our research.

The main output from this study will be evidence on whether our COVID-19-specific clinical nursing care protocol delivers additional benefits in overall patient experience, care quality, functional ability and treatment outcomes, with cost-effectiveness estimates, for hospitalised patients with the SARS-CoV-2 virus not invasively ventilated, compared with care as usual in hospital wards. We will have a programme theory allowing us to potentially generalise any such benefits to environments like care homes, patients with other conditions requiring isolation and to global health systems. If effective, our guidelines, education materials and strategies will be made accessible via Health Education England, the Open University's FutureLearn[38] web platform and through experienced NHS sites acting as training and dissemination hubs for other health Trusts, environments such as care homes and global health systems.

## DISCUSSION

The strengths of this study include that it is the first randomised controlled trial of a fundamental care clinical nursing protocol for patients infected with the SARS-CoV-2 virus and the intervention relates specifically to these patients admitted to inpatient wards who are not invasively ventilated. Our intervention programme theory will enable generalisation of our findings to other environments such as care homes, patients with other conditions requiring isolation and global health systems. In terms of limitations, the trial coprimary outcomes require patient participants to have capacity to consent and report their experience of care, and, therefore, we will be unable to collect and report data from patients who lack this capacity. In common with other cluster randomised controlled trials of behaviour change interventions, we are unable to blind nurses, participants or data collectors to trial arm allocation.

Our research aligns with the WHO COVID-19 R&D Roadmap priorities and research gaps,[70] viz: 'to determine optimal clinical practice strategies to improve the processes of care'. It includes 'rapid approaches to capture healthcare worker views (surveys, interviews)' and a rapid ethnograph(y) in healthcare'.[70] It will, therefore, provide guidance to nurses attempting to overcome significant barriers to providing fundamental physical, relational and psychosocial care to patients, specifically those with the SARS-CoV-2 virus not invasively ventilated but with a generalisable programme theory suitable for adaptation to other care environments, other pandemics and other nations globally.

**Author affiliations**
[1]Institute for Health Research, College of Medicine and Health, University of Exeter, Exeter, UK
[2]Department of Health and Caring Sciences, Western Norway University of Applied Sciences, Bergen, Norway
[3]Nottingham University Hospitals NHS Trust, Nottingham, UK
[4]Royal Marsden NHS Foundation Trust, London, UK
[5]Community Health Sciences, University of Nottingham, Nottingham, UK
[6]University of Salford, Salford, UK
[7]Florence Nightingale School of Nursing and Midwifery, King's College London, London, UK
[8]Academy of Nursing, College of Medicine and Health, University of Exeter, Exeter, UK
[9]NIHR Clinical Research Facility, University of Exeter, Exeter, UK
[10]Cardiac/Pulmonary Rehabilitation, Leicester Royal Infirmary, Leicester, UK
[11]Insitute of Human Nutrition, University of Southampton, Southampton, UK
[12]Clinical Trials Unit, University of Exeter, Exeter, UK

**Contributors** DR wrote the protocol with input from co-investigators HVRS, EC, JC, SC, CH, PL, HI-S, GJM-T, NR, AMRa, AMRu, MS, SJS, J, T-C, ST and SW. All other authors (RA, AB, SC, FD, LQ, HT, FW, RW, JB, HAH, KM, LM, NM and LR) reviewed the manuscript and made editorial suggestions. FD and NR are independent patient and public Involvement members of the team, GJM-T is the senior trial statistician and CH the trial economist. The trial team include nurses (DR, JC, SC, HI-S, GJM-T, AMRa, AMRu, MS, ST, LM, NM, LR), a physiotherapist (SJS), an occupational therapist (PL), a dietician (SW), medical statisticians (GJM-T, SC, FCW), an information specialist (AB), systematic reviewers (JT-C, RA, RW), a health economist (CH), health scientists and clinical trialists (DAR, HVRS, LQ, HT, JB, HAH), patient and public involvement specialists (EC, FD, NR) and an administrator (MK).

**Funding** This work is supported by National Institute for Health Research and UK Research and Innovation, administered by the MRC: grant number MR/V02776X/1. The trial sponsor is the University of Exeter, ref: 1920/Research Ethics and Governance Office, Lafrowda House, St Germans Road, Exeter, Devon, EX4 6TL.

**Disclaimer** Neither the funder nor sponsor will have any role or ultimate authority in study design; collection, management, analysis, and interpretation of data; writing of the report; and the decision to submit the report for publication.

**Competing interests** None declared.

**Patient consent for publication** Not required.

**Provenance and peer review** Not commissioned; externally peer reviewed.

**ORCID iDs**
David A Richards http://orcid.org/0000-0002-8821-5027
Anne-Marie Russell http://orcid.org/0000-0002-0468-3537
Maggie Shepherd http://orcid.org/0000-0003-2660-0955
Jo Thompson Coon http://orcid.org/0000-0002-5161-0234
Rebecca Abbott http://orcid.org/0000-0003-4165-4484
Harriet A Hunt http://orcid.org/0000-0003-1254-0568

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
