## [Reviewer comments · BMJ Open]

ARTICLE DETAILS

TITLE (PROVISIONAL)	COVID-NURSE: evaluation of a fundamental nursing care protocol compared to care as usual on experience of care for non-invasively ventilated patients in hospital with the SARS-CoV-2 virus: protocol for a cluster randomised controlled trial.
AUTHORS	Richards, David; Rose Sugg, Holly; Cockcroft, Emma; Cooper, Joanne; Cruickshank, Susanne; Doris, Faye; Hulme, Claire; Logan, Phillipa; Iles-Smith, Heather; Melendez-Torres, G.J; Rafferty, Anne Marie; Reed, Nigel; Russell, Anne-Marie; Shepherd, Maggie; Singh, Sally; Thompson Coon, Jo; Tooze, Susannah; Wootton, Stephen; Abbott, Rebecca; Bethel, Alison; Creanor, Siobhan; Quinn, Lynne; Tripp, Harry; Warren, Fiona; Whear, Rebecca; Bollen, Jessica; Hunt, Harriet; Kent, Merryn; Morgan, Leila; Morley, Naomi; Romanczuk, Lidia

VERSION 1 – REVIEW

REVIEWER	Hosie, Annmarie The University of Notre Dame Australia, School of Nursing Sydney
REVIEW RETURNED	12-Dec-2020

GENERAL COMMENTS	Thank you for the opportunity to review your manuscript of a protocol of a cluster RCT of nursing intervention for patients in the context of COVID-19. I was very pleased to read of this excellent trial initiative and firstly would like to congratulate the team for their leadership. The manuscript is well written and structured in many respects. However, there are aspects requiring attention before it is ready for publication. These are as follows: Major 1. A clearer explanation and presentation of the study design is required. The design is stated to be a rapid-cycle cluster RCT, which should be explained/defined as not all readers will be familiar with it. However, towards the end of the manuscript, it is further stated that the trial is a combined effectiveness/implementation trial and that it is underpinned by program logic. While none of these statements are contradictory, the latter terms/points should be presented and defined in the design section. Importantly, given the complexity of the trial design, a schematic diagram would be helpful for the readers (and is also highly recommended in the SPIRIT checklist). A logic model (as is customary in studies/projects using program theory) would also be helpful. Lastly regarding design, you state that a hierarchy of outcomes was not necessary to specify because the trial is a combined effectiveness/implementation, yet guidance for a
--

hierarchy of outcomes in such types of designs has been outlined by Curran et al (see <https://pubmed.ncbi.nlm.nih.gov/22310560/>). In that paper, three types of hybrid effectiveness/implementation trials and the corresponding outcome foci are presented. My suggestion is that the terminology and typology outlined by Curran would also inform and clarify the explanation of your study design.

2. Related to the above points, I found the separation of process and economic outcomes from patient outcomes to be puzzling. I believe it would be clearer to the reader if all outcomes are presented together (although they still can be categorised in the same way). To this end, the RE-AIMS outcomes framework for effectiveness/implementation studies may be useful in the presentation of the outcomes outlined in your protocol. See <https://www.re-aim.org/about/what-is-re-aim/>

3. Intervention: more details on the strategies for fundamental care are required. Can the guideline be presented as a supplementary file? As without these details, the reader cannot tell exactly what it is that nurses will be taught/encouraged to do at the intervention sites.

4. A schedule of study measures and timepoints is customary to provide in a trial protocol, and would help to clarify the data collection points in the trial. On this note, I also wondered about the ceiling for data collection from patients - i.e. the minimum time point of receipt of nursing care is 72 hours, but what is the maximum?

Minor

5. What are the dates for the trial? I.e. has it started yet, and if not, when is the planned starting date?

6. How many participants (patients and nurses) will be recruited?

7. How will you deal with missing data? As some patients with COVID-19 will unfortunately die, and many will experience delirium, meaning that not all who are recruited will be able to complete all of the study measures.

8. Please provide a reference for the Caldicott Principles, given the international readership.

9. Why is the recording and reporting of non-serious AEs not required in your trial? Please explain why.

10. Who will obtain consent from participants?

11. Please provide a reference for the WHO COVID-19 R&D.

12. The SPIRIT checklist item 32 refers to sharing of the consent form and other related documents for participants - why have you stated this is not applicable for your trial?

13. There are a couple of overly long sentences that are hard to follow. Please revise these in the Methods and Analysis section of the abstract, and the second last paragraph of the Background section.

14. Please acknowledge the lack of blinding of nurses, patients, and data collectors in the limitations section. Another limitation is that by requiring patients to be capable of providing informed consent, those with impaired decision-making capacity (such as from dementia) will not be included, which means that the sample will not be representative of overall patient population likely to need care due to COVID-19.

15. The process and economic outcomes are not outlined in the Abstract.

16. Check reference 44 - the first word of the article title is missing a capital letter.

17. SPIRIT checklist item 5d recommends that the composition, roles, and responsibilities of the trial team should be described. It wasn't clear to me what the investigators' disciplines are, which I suggest is an important aspect to describe given that the

	intervention is focused on nursing care. Thank you again for undertaking this important study.
--	---

REVIEWER	Couper, Keith University of Warwick, Warwick Medical School
REVIEW RETURNED	18-Jan-2021

GENERAL COMMENTS	Thank you for submitting this trial protocol to BMJ Open. The trial is extremely important, and the publication of the trial protocol is to be welcomed. I appreciate that this is a complex trial with several components. From reading the protocol paper, I felt that the reader would benefit from a fuller description of some components. I have a number of suggestions that I hope will improve the clarity of the protocol paper:  - My understanding is that randomisation occurs at the hospital-level. In intervention hospitals, is the intervention delivered to all wards, targeted at wards caring for patients with COVID-19, or a single COVID-19 ward at each hospital? - My understanding of your randomisation process is that you will match two hospitals based on two characteristics, and randomise one to intervention and one to control. This could potentially lead to recruitment of 18 hospitals with similar characteristics (e.g. research intensive with low diversity)- was this considered at the design stage? - A key determinant of quality of care relates to staff level of education, patient acuity, and nurse: patient ratio. The impact of COVID-19 on individual hospitals is obviously unpredictable, but it seems there is a risk that these factors might easily be unbalanced across study hospitals (particularly as the number of clusters is low). Have you considered how these factors might be addressed at the analysis stage? - It would be helpful to provide more clarity about the three-wave concept. How long does each wave last and how long is each review process? Will participants be recruited during each review process? - A key issue is that over 20-weeks across 18 hospitals, far more than 1080 patients will be treated with COVID-19. This seems to create a significant issue with selection bias as research teams will not be blinded and may preferentially choose patients based on who they presume has been satisfied (or perhaps not satisfied) with their care. It would be helpful if the authors could comment on this. - What screening data will be collected? - If the target recruitment is achieved in fewer than 20-weeks, will the trial stop or will it over-recruit? - What is the target sample size for health professionals? - The description of the interventions is rather vague. My understanding is that the trial has started, so a clearer description of interventions should be possible. The TIDIER checklist developed by Hoffmann and colleagues may provide a helpful format (doi: https://doi.org/10.1136/bmj.g1687) - In your trial registration, it suggests that EQ-5D will be collected at a single time point (>72-hours). In the protocol, it refers to collection at baseline and discharge- please clarify (and update where needed). - Will the analysis account for the timepoint at which outcomes are collected? My understanding is that outcomes may be collected at any point between 72-hours and discharge- is there a risk that the outcome collection timepoint could systematically differ between
---

	hospitals?  - How will you collect the relevant cost data (e.g. ICU length of stay)? Will you also collect data on higher-cost drugs such as remdesivir? - It would be helpful to include the full protocol as a supplementary appendix. - Please confirm date of trial registration, and whether it was prospectively or retrospectively registered.
--	---

REVIEWER	Kuhn, Lisa Monash University School of Nursing and Midwifery
REVIEW RETURNED	07-Feb-2021

GENERAL COMMENTS	Dear Dr Sucksmith, Thank you for the opportunity to review this manuscript entitled "COVID-NURSE: development and evaluation of the effects of a COVID-specific fundamental nursing care protocol compared to care as usual on experience of care for non-invasively ventilated patients in hospital with the SARS-CoV-2 virus: protocol for a cluster randomised controlled trial." The protocol for the proposed trial is registered, funded and Ethics' approved. Overview The paper is topical and I believe will likely be of interest to BMJ Open's readership with revision, some more fine-tuning and detail. I have not seen a similar protocol for proposed research in the literature elsewhere since onset of the COVID-19 pandemic. The authors, who are not blinded to me, provide an extensive discussion about the planning and rationale for a comprehensive nursing-centred cluster randomised study regarding COVID-nursing care in non-invasively ventilated hospital patients in the United Kingdom. There are few if any typographical errors throughout the paper of note and the referencing is consistent and contemporary. NB: There are no authors listed for the PROSPERO study at reference No. 23. My overall feeling upon reading the protocol several times is that the authors have likely spent so much time on it that they have assumed they have explicated some of the background to the trial's rationale and what their intervention will be, that has led to some of these details being overlooked. There was considerable assumed knowledge such as the concept of fundamental nursing care comprising relational and transactional behaviours, but what the focus of the intervention was going to be escaped me: apart from wearing PPE, what will the COVID-specific fundamental nursing care entail? How will this be demonstrably different than the 'usual' fundamental nursing care? What if the nurses practice these same behaviours in the usual (control) arm? The three-wave cluster randomised controlled trial (RCT) proposed is complicated (reasonably so) and would benefit from some sort of graphical representation to illustrate clearly how this will unfold. It might assist the reader if it was set out in a Gantt chart. I did not find a clear hypothesis or research question or aim set out in the work, so I cannot assess easily how these will be answered. I would need more detail about timelines and the actual nursing care to be evaluated to understand if the trial would be feasible and without any instruments/tools or variables or indication of the nursing transactional (+/-) relational behaviours that will be the intervention, I do not believe the study would be reproducible from reading this protocol.
---

	Other than these points, my other thoughts on sections of the paper are outlined below, as they appear: Abstract P. 3, Line 27. How is service use correlated with patient experience? I suggest removing 'service use' from this sentence. Line 31. The sentence could read: "The high risk of transmission of SARS-CoV-2 and measures to reduce this such as personal protective equipment (PPE) compromise nurses' relational and transactional care". Methods and analysis. So, the co-primary outcomes are completion of two instruments in their entirety? Line 46. Suggest changing to "secondary outcomes will include measures of quality of care (e.g. pressure sores, falls, medication errors)..." Line 53. The sentence starting with "We will estimate..." is very dense and might be better broken into parts to allow the reader to digest. Line 58. Sub-group analysis will be conducted to compare the impact of ethnicity on outcomes, but it needs to be included somewhere in the paper why ethnicity is important (compared to age and gender for instance) and does this mean ethnicity of the patients or the nurses or both? P. 4. Strengths and limitations of the study. Only strengths are included here. Also, is the intervention programme theory the intervention theory? A reference would help. Background. Again, I am unsure how overall service use is affected by patients' experiences of nursing care. Does it mean that they just do not return or refuse to be admitted if they experience poor service? Is relational care necessary to meet a patient's fundamental care needs? It is not clear here. The explanation of the PROMs and PREMs was useful. P. 5. The paragraph starting with "Currently, no international..." needs more depth and evidence to support it. Why does variation in patient experience affect patient outcomes? Where is the evidence that nurses, teams and organisations have rapidly adapted to COVID-19 and how this adaptation compares to the 'usual nursing care'? Is the aim of the research covered in the part of the sentence "we do not know how to optimise nursing care for the specific challenges of nursing COVID-19 patients..."? The paragraph starting with "Between July and October..." includes an outline of the protocol's development for clinical nursing is comprehensive, but I am unsure what the protocol to be compared to usual care includes. I went back to the journal's reviewer page in case I had missed a summary of what was going to be measured against usual care, but did not find one and think it would be useful in the protocol. What was the consensus opinion reached? What was the evidence-based nursing protocol (Line 50)? The references 24 and 25 are included as though they were part of the survey re COVID-19 care, but they were published in 2014 and 2018. P. 6, Line 9: Does setting type mean urban/regional/rural or how the service is funded? Line 16. What are "nursing care workers"? Are they nurses? Can they be expected to provide equivalent care to a registered nurse? Line 25. "care as usual plus the clinical protocol developed as described above". What was the clinical protocol developed that I should follow if I was a nurse working in the intervention group? Line 37. How will "Leadership from ward managers and senior nurses" be different than usual care and how will this be measured?
--	---

	Line 53. How will you measure the non-technical/procedural nursing care? P. 7, Line 47. Change to: “We will use these data to...” P. 8, Line 58. Caldicott Principles needs a reference. P. 9, Line 20. Will sub-group analyses also be conducted for other demographic differences (e.g. gender and age)? Was there a reason to only look at ethnicity? Line 28. As mentioned earlier, care staff members’ roles, responsibilities need to be provided for the audience. Presumably they are different to registered nurses (RNs)? The international audience may not understand the nuances or whether they can be compared with RNs. Line 58. Is the Framework analysis reference incorporated into reference No. 52 cited here? P. 10, Line 40. Is there a risk of bias (selective reporting) if only some SAEs are reported and only if the CI deems them to be relevant? P. 11, Line 8. Suggest to change to “responsive to participant needs.” Line 20. Suggest to change to “from participants who individually consent.” Lines 26-29. Repeated from earlier. Line 55. Should this be well developed NHS sites rather than through “experienced...” P. 12. Line 5. WHO COVID-19 R&D Roadmap needs a reference. Again, thank you for asking me to review this paper. In summary, I do have some concerns in terms of stated hypothesis, research question and/or aims. Without seeing the “COVID-specific clinical nursing care protocol” (p. 11, Line 43) referred to, I cannot evaluate whether this protocol is likely to be feasible or reproducible, as I would normally expect to when reading a protocol. With revision of the paper to include the above-mentioned components, the proposed protocol would likely be of interest to BMJ Open’s readers.
--	---

VERSION 1 – AUTHOR RESPONSE

Reviewer: 1

Dr. Annmarie Hosie, The University of Notre Dame Australia

Comments to the Author:

Thank you for the opportunity to review your manuscript of a protocol of a cluster RCT of nursing intervention for patients in the context of COVID-19.

I was very pleased to read of this excellent trial initiative and firstly would like to congratulate the team for their leadership.

The manuscript is well written and structured in many respects. However, there are aspects requiring attention before it is ready for publication.

These are as follows:

Major

1. A clearer explanation and presentation of the study design is required. The design is stated to be a rapid-cycle cluster RCT, which should be explained/defined as not all readers will be familiar with it. However, towards the end of the manuscript, it is further stated that the trial is a combined effectiveness/implementation trial and that it is

underpinned by program logic. While none of these statements are contradictory, the latter terms/points should be presented and defined in the design section. Importantly, given the complexity of the trial design, a schematic diagram would be helpful for the readers (and is also highly recommended in the SPIRIT checklist). A logic model (as is customary in studies/projects using program theory) would also be helpful. Lastly regarding design, you state that a hierarchy of outcomes was not necessary to specify because the trial is a combined effectiveness/implementation, yet guidance for a hierarchy of outcomes in such types of designs has been outlined by Curran et al (see <https://eur03.safelinks.protection.outlook.com/?url=https%3A%2F%2Fpubmed.ncbi.nlm.nih.gov%2F22310560%2F&data=04%7C01%7CD.A.Richards%40exeter.ac.uk%7C4528ccab045141a9f96608d8cc4c5868%7C912a5d77fb984eeef321334d8f04a53%7C0%7C0%7C637483975584784701%7CUnknown%7CTWFpbGZsb3d8eyJWljojMC4wLjAwMDAiLCJQljojV2luMzliLCJBTil6lk1haWwiLCJXVCI6Mn0%3D%7C1000&reserved=0>). In that paper, three types of hybrid effectiveness/implementation trials and the corresponding outcome foci are presented. My suggestion is that the terminology and typology outlined by Curran would also inform and clarify the explanation of your study design.

NOTWITHSTANDING THE DESIGN NOW HAS CHANGED, WE APPRECIATE THE REVIEWER'S COMMENTS AND HAVE EXPLAINED THE ORIGINAL RAPID CYCLE DESIGN MORE CLEARLY.

ON REFLECTION AND AS A CONSEQUENCE OF THIS DESIGN CHANGE, WE HAVE REMOVED REFERENCE TO THE TRIAL BEING A COMBINED EFFECTIVENESS AND IMPLEMENTATION ONE AS WE THINK THIS IS CONFUSING AND PROBABLY NOT QUITE ACCURATE. WE HAVE USED LESSONS LEARNED FROM IMPLEMENTATION SCIENCE TRIALS, BUT UNLIKE THESE KINDS OF TRIAL WHERE THE FOCUS IS ON INVESTIGATING THE EFFECTIVENESS OF STRATEGIES TO IMPLEMENT KNOWN CLINICALLY EFFECTIVE INTERVENTIONS, WE ARE INVESTIGATING CLINICAL EFFECTIVENESS. WE ARE USING PROCESS EVALUATION TECHNIQUES TO HELP US UNDERSTAND FIDELITY TO THE INTERVENTION, BUT THIS IS NOT THE SAME AS A TRIAL OF INTERVENTION TECHNIQUES *PER SE*. AS THE REVIEWER WILL NOTE, OUR ANALYSIS PLAN HAS CHANGED IN LINE WITH OUR MAJOR DESIGN CHANGE ANYWAY, AND THIS IS DESCRIBED FULLY.

2. Related to the above points, I found the separation of process and economic outcomes from patient outcomes to be puzzling. I believe it would be clearer to the reader if all outcomes are presented together (although they still can be categorised in the same way). To this end, the RE-AIMS outcomes framework for effectiveness/implementation studies may be useful in the presentation of the outcomes outlined in your protocol. See <https://eur03.safelinks.protection.outlook.com/?url=https%3A%2F%2Fwww.re-aim.org%2Fabout%2Fwhat-is-re-aim%2F&data=04%7C01%7CD.A.Richards%40exeter.ac.uk%7C4528ccab045141a9f96608d8cc4c5868%7C912a5d77fb984eeef321334d8f04a53%7C0%7C0%7C637483975584784701%7CUnknown%7CTWFpbGZsb3d8eyJWljojMC4wLjAwMDAiLCJQljojV2luMzliLCJBTil6lk1haWwiLCJXVCI6Mn0%3D%7C1000&reserved=0>

THIS POINT IS RELATED TO THAT ABOVE AND IS A DECISION THAT ALL TRIAL INVESTIGATORS MUST MAKE AS DESCRIBED BY THE MRC'S GUIDANCE – THE LEVEL OF SEPARATION AND INTEGRATION OF PROCESS DATA WITH OUTCOME DATA. SEPARATION IS ONE OF THE DESIGN AND ANALYSIS OPTIONS SUGGESTED BY MRC GUIDANCE. IN MANY CASES PROCESS EVALUATION PROTOCOLS ARE PUBLISHED SEPARATELY AND BY PUBLISHING THIS TOGETHER WITHIN THE MAIN OUTCOMES PAPER WE RECOGNISE THIS MAY APPEAR CONFUSING. NONETHELESS, WE PREFER TO KEEP THEM APART IN THE PAPER FOR CONCEPTUAL REASONS, AND NOT LEAST BECAUSE PROCESS DATA ANALYSTS ARE NECESSARILY UNBLINDED TO ALLOCATION

AND WE FEEL IT IS BETTER TO MAINTAIN CLARITY BY HAVING PROCESS DATA IN A SEPARATE SECTION.

3. Intervention: more details on the strategies for fundamental care are required. Can the guideline be presented as a supplementary file? As without these details, the reader cannot tell exactly what it is that nurses will be taught/encouraged to do at the intervention sites.

WE ABSOLUTELY UNDERSTAND THIS POINT, WHICH HAS BEEN SUGGESTED BY MORE THAN ONE REVIEWER, AND INDEED WE WISH WE COULD PUBLISH THE CLINICAL PROTOCOL. HOWEVER, TO DO SO WOULD UNBLIND CURRENT AND FUTURE CONTROL SITES ALLOCATED TO THE USUAL CARE ARM OF THE TRIAL. RELUCTANTLY, THEREFORE, WE CANNOT PROVIDE THIS DETAIL.

4. A schedule of study measures and timepoints is customary to provide in a trial protocol, and would help to clarify the data collection points in the trial. On this note, I also wondered about the ceiling for data collection from patients - i.e. the minimum time point of receipt of nursing care is 72 hours, but what is the maximum?

WE HAVE CLARIFIED THIS POINT TO MAKE IT CLEAR THAT WE CAN COLLECT DATA DURING ADMISSION OR A MAXIMUM OF TWO WEEKS POST DISCHARGE

Minor

5. What are the dates for the trial? I.e. has it started yet, and if not, when is the planned starting date?

WE HAVE ADDED THIS DETAIL TO THE DESIGN SECTION

6. How many participants (patients and nurses) will be recruited?

WE HAVE ADDED THIS TO THE REVISED SAMPLE SIZE AND POWER CALCULATION SECTION

7. How will you deal with missing data? As some patients with COVID-19 will unfortunately die, and many will experience delirium, meaning that not all who are recruited will be able to complete all of the study measures.

WE HAVE ADDED A SECTION ON MANAGEMENT OF MISSING DATA.

8. Please provide a reference for the Caldicott Principles, given the international readership

WE HAVE DONE THIS

9. Why is the recording and reporting of non-serious AEs not required in your trial? Please explain why.

WE HAVE EXPLAINED THIS FULLY WITH REFERENCE TO CURRENT CLINICAL TRIALS LEGISLATION AND REGULATION

10. Who will obtain consent from participants?

WE HAVE CLARIFIED THIS IN THE SECTION 'DATA COLLECTION' AND PROVIDED THE RELEVANT REFERENCE.

11. Please provide a reference for the WHO COVID-19 R&D

ADDED: (THIS WAS INCLUDED AT THE END OF THE SUBSEQUENT SENTENCE IN THE MANUSCRIPT BUT IS REPLICATED HERE FOR CLARITY)

12. The SPIRIT checklist item 32 refers to sharing of the consent form and other related documents for participants - why have you stated this is not applicable for your trial?

THIS WAS AN ERROR. WE HAVE ADDED IT

13. There are a couple of overly long sentences that are hard to follow. Please revise these in the Methods and Analysis section of the abstract, and the second last paragraph of the Background section.

WE HAVE AMENDED THESE PARAGRAPHS AS SUGGESTED

14. Please acknowledge the lack of blinding of nurses, patients, and data collectors in the limitations section. Another limitation is that by requiring patients to be capable of providing informed consent, those with impaired decision-making capacity (such as from dementia) will not be included, which means that the sample will not be representative of overall patient population likely to need care due to COVID-19.

WE HAVE ADDED THESE POINTS TO THE STRENGTHS AND LIMITATIONS SECTION

15. The process and economic outcomes are not outlined in the Abstract.

WE ARE LIMITED SOMEWHAT BY WORD COUNT BUT HAVE AMENDED

16. Check reference 44 - the first word of the article title is missing a capital letter.

CORRECTED

17. SPIRIT checklist item 5d recommends that the composition, roles, and responsibilities of the trial team should be described. It wasn't clear to me what the investigators' disciplines are, which I suggest is an important aspect to describe given that the intervention is focused on nursing care.

WE HAVE ADDED THIS DETAIL TO THE AUTHORS' CONTRIBUTIONS SECTION

Thank you again for undertaking this important study.

Reviewer: 2

Dr. Keith Couper, University of Warwick

Comments to the Author:

Thank you for submitting this trial protocol to BMJ Open.

The trial is extremely important, and the publication of the trial protocol is to be welcomed.

I appreciate that this is a complex trial with several components. From reading the protocol paper, I felt that the reader would benefit from a fuller description of some components. I have a number of suggestions that I hope will improve the clarity of the protocol paper:

- My understanding is that randomisation occurs at the hospital-level. In intervention hospitals, is the intervention delivered to all wards, targeted at wards caring for patients with COVID-19, or a single COVID-19 ward at each hospital

WE HAVE ADDED THE FOLLOWING STATEMENT TO THE SECTION ON PATIENT ELIGIBILITY: 'WE WILL RECRUIT PARTICIPANTS SPECIFICALLY FROM WARDS ALLOCATED TO THE CARE OF PATIENTS ADMITTED FOR TREATMENT OF COVID-19 SYMPTOMS.'

- My understanding of your randomisation process is that you will match two hospitals based on two characteristics, and randomise one to intervention and one to control. This could potentially lead to recruitment of 18 hospitals with similar characteristics (e.g. research intensive with low diversity)- was this considered at the design stage?

WE BELIEVE THAT THIS QUESTION HAS BEEN SUPERCEDED BY OUR DESIGN CHANGE. RECRUITMENT OF SITES HAS BEEN SO DIFFICULT THAT WE DID NOT HAVE THE ABILITY TO PICK AND CHOOSE BASED ON THE DISTRIBUTION OF THESE CHARACTERISTICS OF SITES. WE WILL NOW CONTROL FOR THESE IN OUR ANALYSES

- A key determinant of quality of care relates to staff level of education, patient acuity, and nurse: patient ratio. The impact of COVID-19 on individual hospitals is obviously unpredictable, but it seems there is a risk that these factors might easily be unbalanced across study hospitals (particularly as the number of clusters is low). Have you considered how these factors might be addressed at the analysis stage?

WE ARE COLLECTING THIS DATA AS PART OF OUR PROCESS EVALUATION

- It would be helpful to provide more clarity about the three-wave concept. How long does each wave last and how long is each review process? Will participants be recruited during each review process?

AS NOTED ABOVE, THE DESIGN HAS NOW CHANGED

- A key issue is that over 20-weeks across 18 hospitals, far more than 1080 patients will be treated with COVID-19. This seems to create a significant issue with selection bias as research teams will not be blinded and may preferentially choose patients based on who they presume has been satisfied (or perhaps not satisfied) with their care. It would be helpful if the authors could comment on this.

DATA WILL BE COLLECTED BY RESEARCH STAFF WHO ARE NOT FAMILIAR WITH THE PATIENTS. THEY WILL APPROACH ALL POTENTIAL PARTICIPANTS AND WE WILL KEEP A LOG FOR THE CONSORT DIAGRAM ON HOW MANY HAVE BEEN APPROACHED. IT IS NOT OF COURSE TOTALLY POSSIBLE TO CONTROL FOR THIS EVENTUALITY BUT WE BELIEVE IT IS AN ISSUE ACROSS ALL TRIALS AND ESPECIALLY THOSE OF BEHAVIOURAL CHANGE INTERVENTIONS. LOCAL TRIAL MANAGEMENT MONITORING PROCEDURES WILL ENSURE THAT RESEARCH STAFF ADHERE TO GCP PRINCIPLES

- What screening data will be collected?

WE ARE NOT SURE TO WHAT THIS QUESTION REFERS. WE WILL COLLECT NUMBERS OF PATIENT AND NURSE PARTICIPANTS APPROACHED AS IS USUAL FOR THE CONSORT DIAGRAM BUT NO OTHER SCREENING DATA

- If the target recruitment is achieved in fewer than 20-weeks, will the trial stop or will it over-recruit?

WE WILL CONTINUE UNTIL WE HAVE COLLECTED SUFFICIENT PARTICIPANTS FROM EACH CLUSTER. SOME CLUSTERS MAY OVERRECRUIT INDIVIDUALS, BUT THE KEY DETERMINANT IN A CLUSTER TRIAL IS CLUSTER NUMBER (WHETHER FROM THE ORIGINAL MATCHED PAIRS DESIGN OR THE REVISED SIMPLE CLUSTER TRIAL) SO THERE IS THE POSSIBILITY THAT WE MAY OVERRECRUIT, WITHOUT COST OR PREJUDICE TO THE ANALYSIS.

- What is the target sample size for health professionals?

THE TRIAL IS NOT POWERED ON THIS POPULATION. WE AIM TO RECRUIT AS MANY NURSES AS WE CAN WHO HAVE CARED FOR PATIENT PARTICIPANTS IN EACH SITE. THIS SITUATION IS EXTREMELY FLUID, IN THAT SOME NURSES REMAIN IN TIGHTLY KNIT COVID TEAMS, WHILE IN OTHER HOSPITALS NURSES MOVE IN AND OUT OF SUCH TEAMS RAPIDLY. IN CONSULTATION WITH CLINICAL COLLEAGUES IN OUR INVESTIGATOR AND ADVISORY GROUPS WE WERE UNABLE TO PREDICT A STANDARD PATTERN AND SO DID NOT POWER THE TRIAL IN THIS WAY. NOTWITHSTANDING THIS, WE ANTICIPATE AROUND 50 NURSES PER SITE (700 IN TOTAL) WILL CONSENT TO GIVE US DATA AND AS SUCH THIS IS OUR OPERATIONAL NUMBER. WE HAVE ADDED THIS FIGURE TO THE TEXT.

- The description of the interventions is rather vague. My understanding is that the trial has started, so a clearer description of interventions should be possible. The TIDIER checklist developed by Hoffmann and colleagues may provide a helpful format

(doi: [https://eur03.safelinks.protection.outlook.com/?url=https%3A%2F%2Fdoi.org%2F10.1136%2Fbmj.g1687&data=04%7C01%7CD.A.Richards%40exeter.ac.uk%7C4528ccab045141a9f96608d8c4c5868%7C912a5d77fb984eeef321334d8f04a53%7C0%7C0%7C637483975584784701%7CUnkown%7CTWFpbGZsb3d8eyJWljoImMC4wLjAwMDAiLCJQIjoiV2luMzliLCJBTiI6IjEhaWwiLCJXVCi6Mn0%3D%7C1000&data=04%7C01%7CD.A.Richards%40exeter.ac.uk%7C4528ccab045141a9f96608d8c4c5868%7C912a5d77fb984eeef321334d8f04a53%7C0%7C0%7C637483975584784701%7CUnkown%7CTWFpbGZsb3d8eyJWljoImMC4wLjAwMDAiLCJQIjoiV2luMzliLCJBTiI6IjEhaWwiLCJXVCi6Mn0%3D%7C1000&reserved=0](https://eur03.safelinks.protection.outlook.com/?url=https%3A%2F%2Fdoi.org%2F10.1136%2Fbmj.g1687&data=04%7C01%7CD.A.Richards%40exeter.ac.uk%7C4528ccab045141a9f96608d8c4c5868%7C912a5d77fb984eeef321334d8f04a53%7C0%7C0%7C637483975584784701%7CUnkown%7CTWFpbGZsb3d8eyJWljoImMC4wLjAwMDAiLCJQIjoiV2luMzliLCJBTiI6IjEhaWwiLCJXVCi6Mn0%3D%7C1000&data=04%7C01%7CD.A.Richards%40exeter.ac.uk%7C4528ccab045141a9f96608d8c4c5868%7C912a5d77fb984eeef321334d8f04a53%7C0%7C0%7C637483975584784701%7CUnkown%7CTWFpbGZsb3d8eyJWljoImMC4wLjAwMDAiLCJQIjoiV2luMzliLCJBTiI6IjEhaWwiLCJXVCi6Mn0%3D%7C1000&data=04%7C01%7CD.A.Richards%40exeter.ac.uk%7C4528ccab045141a9f96608d8c4c5868%7C912a5d77fb984eeef321334d8f04a53%7C0%7C0%7C637483975584784701%7CUnkown%7CTWFpbGZsb3d8eyJWljoImMC4wLjAwMDAiLCJQIjoiV2luMzliLCJBTiI6IjEhaWwiLCJXVCi6Mn0%3D%7C1000&reserved=0))

AS NOTED IN OUR RESPONSE TO THIS SAME POINT FROM REVIEWER 1, WE ABSOLUTELY UNDERSTAND THIS POINT, AND INDEED WE WISH WE COULD PUBLISH THE CLINICAL PROTOCOL. HOWEVER, TO DO SO WOULD UNBLIND CURRENT AND FUTURE CONTROL SITES ALLOCATED TO THE USUAL CARE ARM OF THE TRIAL. RELUCTANTLY, THEREFORE, WE CANNOT PROVIDE THIS DETAIL.

- In your trial registration, it suggests that EQ-5D will be collected at a single time point (>72-hours). In the protocol, it refers to collection at baseline and discharge- please clarify (and update where needed).

THANK YOU, THIS WAS A MISTAKE CARRIED OVER FROM PREVIOUS DRAFTS. WE WILL ONLY COLLECT THIS MEASURE ONCE. THIS HAS BEEN CORRECTED IN THE MANUSCRIPT

- Will the analysis account for the timepoint at which outcomes are collected? My understanding is that outcomes may be collected at any point between 72-hours and discharge- is there a risk that the outcome collection timepoint could systematically differ between hospitals?

THIS IS A PRACTICAL ISSUE ASSOCIATED WITH THE PRESSURES PUT ONTO RESEARCH TEAMS. IDEALLY, WE WOULD WISH TO KEEP THIS IDENTICAL BUT WE RECOGNISE THIS WILL NOT BE POSSIBLE IN PRACTICE. WE ESTABLISHED THE MINIMUM PERIOD OF EXPERIENCE FOR NURSING CARE AS 72 HOURS FROM CONSULTATION WITH OUR PPI COLLEAGUES. CURRENTLY, AVERAGE LENGTH OF STAY IN UK HOSPITALS FOR PATIENTS WITH COVID IS 4-5 DAYS SO THERE IS UNLIKELY TO BE A LARGE DIFFERENCE BETWEEN OUR MINIMUM AND THE ACTUAL LENGTH OF STAY. WE WILL OF COURSE MONITOR THIS.

- How will you collect the relevant cost data (e.g. ICU length of stay)? Will you also collect data on higher-cost drugs such as remdesivir?

WE ARE COLLECTING THIS DATA USING A BESPOKE HOSPITAL CARE USE INVENTORY AND HAVE ADDED REFERENCE TO THIS TO THE ECONOMIC EVALUATION PARAGRAPH

- It would be helpful to include the full protocol as a supplementary appendix.
IF THE REVIEWER MEANS THE CLINICAL PROTOCOL THEN WE REGRET WE ARE UNABLE TO DO THIS. THIS ARTICLE IS AN EXPANDED VERSION OF THE EXPERIMENTAL TRIAL PROTOCOL.
- Please confirm date of trial registration, and whether it was prospectively or retrospectively registered.
WE HAVE ADDED THIS TO THE 'ETHICAL ISSUES' SECTION.

Reviewer: 3

Dr. Lisa Kuhn, Monash University School of Nursing and Midwifery

Comments to the Author:

Dear Dr Sucksmith,

Thank you for the opportunity to review this manuscript entitled "COVID-NURSE: development and evaluation of the effects of a COVID-specific fundamental nursing care protocol compared to care as usual on experience of care for non-invasively ventilated patients in hospital with the SARS-CoV-2 virus: protocol for a cluster randomised controlled trial."

The protocol for the proposed trial is registered, funded and Ethics approved.

Overview

The paper is topical and I believe will likely be of interest to BMJ Open's readership with revision, some more fine-tuning and detail. I have not seen a similar protocol for proposed research in the literature elsewhere since onset of the COVID-19 pandemic. The authors, who are not blinded to me, provide an extensive discussion about the planning and rationale for a comprehensive nursing-centred cluster randomised study regarding COVID-nursing care in non-invasively ventilated hospital patients in the United Kingdom.

There are few if any typographical errors throughout the paper of note and the referencing is consistent and contemporary. NB: There are no authors listed for the PROSPERO study at reference No. 23.

THANK YOU, NOW ADDED

My overall feeling upon reading the protocol several times is that the authors have likely spent so much time on it that they have assumed they have explicated some of the background to the trial's rationale and what their intervention will be, that has led to some of these details being overlooked. There was considerable assumed knowledge such as the concept of fundamental nursing care comprising relational and transactional behaviours, but what the focus of the intervention was going to be escaped me: apart from wearing PPE, what will the COVID-specific fundamental nursing care entail? How will this be demonstrably different than the 'usual' fundamental nursing care? What if the nurses practice these same behaviours in the usual (control) arm?

AS NOTED IN RESPONSE TO SIMILAR ISSUES RAISED BY REVIEWERS 1 AND 2, WE ARE UNABLE TO DESCRIBE THE CLINICAL PROTOCOL IN THE DETAIL REQUESTED, BECAUSE TO DO SO WOULD UNBLIND NURSES WORKING IN CURRENT AND FORTHCOMING PARTICIPATING SITES. OUR DEVELOPMENT WORK INDICATED THAT BARRIERS TO NURSING CARE COMPRISED VERY MUCH MORE THAN THE WEARING OF PPE. WE RECOGNISE THAT READERS AND REVIEWERS WILL NEED TO EXERCISE A DEGREE OF TRUST AND ACCEPT THAT OUR BRIEFLY DESCRIBED THREE PHASE INTERVENTION DEVELOPMENT PROCESS (RAPID REVIEW, SURVEY AND CO-CREATION GROUPS) HAS IDENTIFIED MISSED CARE, BARRIERS AND INNOVATION STRATEGIES, AND THAT THESE HAVE BEEN INCORPORATED INTO THE CLINICAL PROTOCOL. IT IS PRECISELY BECAUSE OF THE POINT RAISED BY THIS REVIEWER – USUAL CARE BEHAVIOURS BEING CONTAMINATED BY EXPOSURE TO THE CLINICAL PROTOCOL IN PRINT – THAT WE ARE UNABLE TO BE MORE EXPLICIT IN THIS MANUSCRIPT.

The three-wave cluster randomised controlled trial (RCT) proposed is complicated (reasonably so) and would benefit from some sort of graphical representation to illustrate clearly how this will unfold. It might assist the reader if it was set out in a Gantt chart.

AS NOTED ABOVE, THE DESIGN HAS NOW CHANGED TO A SIMPLE CLUSTER TRIAL. NONETHELESS, WE HAVE INCLUDED AN EXPLANATION OF WHY WE INITIALLY CHOSE A RAPID CYCLE DESIGN AND WHY THIS PROVED NOT FEASIBLE

I did not find a clear hypothesis or research question or aim set out in the work, so I cannot assess easily how these will be answered.

WE HAVE SEPARATED THIS MORE CLEARLY AT THE END OF THE BACKGROUND SECTION

I would need more detail about timelines and the actual nursing care to be evaluated to understand if the trial would be feasible and without any instruments/tools or variables or indication of the nursing transactional (+/-) relational behaviours that will be the intervention, I do not believe the study would be reproducible from reading this protocol.

WE HAVE REFERRED TO THIS EARLIER AND REGRETABLELY WE DO NOT SEE A WAY TO GIVE FURTHER DETAILS OF THE CLINICAL PROTOCOL AT THIS POINT

Other than these points, my other thoughts on sections of the paper are outlined below, as they appear:

Abstract

P. 3, Line 27. How is service use correlated with patient experience? I suggest removing 'service use' from this sentence.

WE PREFER TO RETAIN THIS AS WE THINK THE LOGIC IS SOUND. PATIENT EXPERIENCE IS ASSOCIATED WITH MISSED CARE. MISSED CARE CAN LEAD TO POOR OUTCOMES AND THEN TO INCREASED READMISSION. WE ARE CAREFUL TO USE THE WORD CORRELATION NOT CAUSE, BUT WE BELIEVE THAT THESE VARIABLES ARE LINKED AND THAT THE REFERENCES CITED SUPPORT THIS

Line 31. The sentence could read: "The high risk of transmission of SARS-CoV-2 and measures to reduce this such as personal protective equipment (PPE) compromise nurses' relational and transactional care".

WE PREFER OUR TEXT. AS NOTED IN OUR ANSWER ABOVE, HIGHLIGHTING PPE AS A DRIVER FOR COMPROMISE IN NURSING CARE, WHILST CERTAINLY AN EXAMPLE, IS BY NO MEANS THE ONLY REASON FOR THIS. WE WANT TO AVOID FOCUS ON ONE PARTICULAR BARRIER TO NURSING CARE WHEN WE KNOW FROM OUR INTERVENTION DEVELOPMENT WORK, INCLUDING OUR SURVEY, THAT OTHER BARRIERS ARE EQUALLY PERTINENT.

Methods and analysis. So, the co-primary outcomes are completion of two instruments in their entirety?

THIS IS CORRECT

Line 46. Suggest changing to "secondary outcomes will include measures of quality of care (e.g. pressure sores, falls, medication errors)..."

THANKS FOR THE SUGGESTION, WE HAVE AMENDED THIS TEXT

Line 53. The sentence starting with "We will estimate..." is very dense and might be better broken into parts to allow the reader to digest.

THANKS FOR THE SUGGESTION, WE HAVE DIVIDED THIS SENTENCE INTO TWO

Line 58. Sub-group analysis will be conducted to compare the impact of ethnicity on outcomes, but it needs to be included somewhere in the paper why ethnicity is important (compared to age and gender for instance) and does this mean ethnicity of the patients or the nurses or both?

WE HAVE ADDED JUSTIFICATION FOR THIS SPECIFIC TO COVID INFECTION AND DEATH RATES WITH A RECENT SYSTEMATIC REVIEW REFERENCE

P. 4. Strengths and limitations of the study. Only strengths are included here. Also, is the intervention programme theory the intervention theory? A reference would help.

WE HAVE ADDED A FEW MORE BALANCED LIMITATIONS. THE STATEMENT RE: PROGRAMME THEORY IS INDEED THE THEORY OF THE INTERVENTION AND WE HAVE ADDED A SENTENCE AND REFERENCES TO THE PROCESS EVALUATION SECTION TO MAKE THIS CLEAR

Background. Again, I am unsure how overall service use is affected by patients' experiences of nursing care. Does it mean that they just do not return or refuse to be admitted if they experience poor service?

ALL OF THE ABOVE COULD BE TRUE, PLUS THE RELATIONSHIPS HIGHLIGHTED IN OUR PREVIOUS RESPONSE TO THIS QUESTION

Is relational care necessary to meet a patient's fundamental care needs? It is not clear here.

THE MODEL (KITSON) WE USE HYPOTHESISES THAT RELATIONAL CARE IS A NECESSARY COMPONENT OF FUNDAMENTAL CARE. THE REVIEWER IS CORRECT IN THAT DEFINITIONS OF FUNDAMENTAL CARE HAVE IN THE PAST BEEN CONTESTED AND HAVE INCLUDED BOTH PATIENT NEEDS AND NURSE ACTIONS. THE CURRENT DEFINITION WE HAVE CITED ACKNOWLEDGES THIS AND INCLUDES CARE ACTIONS AND CARE RECEIPTS. WE CONSIDER THAT MEETING PATIENTS' RELATIONAL NEEDS – INCLUDING ESTABLISHING A RELATIONSHIP WITH PATIENTS AND COMMUNICATING WITH RELATIVES, CARERS AND SIGNIFICANT OTHERS – ARE BOTH ACTIONS AND NEEDS. WE DO NOT DEFINE FUNDAMENTAL CARE NEEDS SOLELY AS PHYSICAL. WE HAVE AMENDED THE TEXT TO ENSURE THIS NUANCE IS CAPTURED

The explanation of the PROMs and PREMs was useful.

THANK YOU

P. 5. The paragraph starting with "Currently, no international..." needs more depth and evidence to support it. Why does variation in patient experience affect patient outcomes? Where is the evidence that nurses, teams and organisations have rapidly adapted to COVID-19 and how this adaptation compares to the 'usual nursing care'? Is the aim of the research covered in the part of the sentence "we do not know how to optimise nursing care for the specific challenges of nursing COVID-19 patients..."?

THE STATEMENT ABOUT NO GUIDANCE IS A STATEMENT OF FACT, AS SHOWN BY OUR SYSTEMATIC REVIEW, WHICH IS BEING SUBMITTED FOR PUBLICATION. WE REFER TO THIS IN THE PROSPERO REGISTRATION. WE HAVE ALSO AMENDED THE PARAGRAPH TO STRESS THE POTENTIAL FOR VARIATION SINCE THERE IS AS YET NO EVIDENCE. NONETHELESS, OUR SURVEY DATA, ALSO REFERRED TO LATER, HAD NURSES REPORTING MISSED CARE FOR PATIENTS WITH COVID-19. TO BE CLEAR, WE MENTION THAT THESE RESULTS WILL BE REPORTED ELSEWHERE AS THERE IS INSUFFICIENT SPACE TO DO SO HERE AND, AS NOTED EARLIER, TO DO SO WOULD RISK UNBLINDING OF PARTICIPANTS IN THE TRIAL. FURTHERMORE, THE LACK OF GUIDANCE AND EVIDENCE OF CONCERN REGARDING MISSED CARE IS INDEED THE RATIONALE FOR UNDERTAKING THIS TRIAL.

The paragraph starting with "Between July and October..." includes an outline of the protocol's development for clinical nursing is comprehensive, but I am unsure what the protocol to be compared to usual care includes. I went back to the journal's reviewer page in case I had missed a summary of what was going to be measured against usual care, but did not find one and think it would be useful in the protocol. What was the consensus opinion reached? What was the evidence-based nursing protocol (Line 50)? The references 24 and 25 are included as though they were part of the survey re COVID-19 care, but they were published in 2014 and 2018.

THIS IS THE SAME PROBLEM REFERRED TO ABOVE. UNTIL THE TRIAL IS COMPLETE WE CANNOT DIVULGE FURTHER DETAILS FOR FEAR OF CONTAMINATING THE USUAL CARE ALLOCATED SITES. REFERENCES 24 AND 25 ARE TO INDICATE WHAT THE DEFINITION OF 'MISSED CARE' IS.

P. 6, Line 9: Does setting type mean urban/regional/rural or how the service is funded?

ALL NHS SERVICES ARE FUNDED THROUGH THE SAME MECHANISM OF TAXATION. THIS IS A STANDARD HEADING THAT MERELY REFERS TO WHAT A CLUSTER IS (IN CONTRAST FOR EXAMPLE TO A PRIMARY CARE PRACTICE)

Line 16. What are “nursing care workers”? Are they nurses? Can they be expected to provide equivalent care to a registered nurse?

THEY ARE NON-REGISTERED STAFF WHO MAKE UP A LARGE PROPORTION OF THE NURSING WORKFORCE IN THE UK AND OTHER COUNTRIES. THEY WORK UNDER THE SUPERVISION OF REGISTERED NURSES. WE HAVE AMENDED THE TEXT ACCORDINGLY

Line 25. “care as usual plus the clinical protocol developed as described above”. What was the clinical protocol developed that I should follow if I was a nurse working in the intervention group?

WE ARE UNABLE TO DESCRIBE THIS IN ANY MORE DETAIL AS STATED EARLIER SINCE TO DO SO WOULD RISK UNBLINDING THE TRIAL. NURSES IN THE INTERVENTION ARM OF THE TRIAL RECEIVE FULL INSTRUCTION AS DESCRIBED LATER

Line 37. How will “Leadership from ward managers and senior nurses” be different than usual care and how will this be measured?

THE GUIDELINE INCLUDES A SECTION FOR NURSE LEADERS SPECIFICALLY RELATING TO THE GUIDELINE CONTENTS

Line 53. How will you measure the non-technical/procedural nursing care?

WE DO NOT UNDERSTAND THIS QUESTION. OUR CO-PRIMARY OUTCOMES ARE BOTH TRANSACTION (QPP) AND RELATIONAL (RACQ) CARE ELEMENTS

P. 7, Line 47. Change to: “We will use these data to...”

THANKS FOR SPOTTING THIS ERROR. CHANGED

P. 8, Line 58. Caldicott Principles needs a reference.

ADDED

P. 9, Line 20. Will sub-group analyses also be conducted for other demographic differences (e.g. gender and age)? Was there a reason to only look at ethnicity?

WE HAVE NO OTHER PLANNED SUB-GROUP ANALYSES. ETHNICITY WAS AN EARLY PREDICTOR OF INCREASED MORTALITY AND MORBIDITY DURING THE FIRST WAVE OF THE PANDEMIC. IT WAS SPECIFICALLY REQUESTED WE EXAMINE THIS BY OUR FUNDER

Line 28. As mentioned earlier, care staff members’ roles, responsibilities need to be provided for the audience. Presumably they are different to registered nurses (RNs)? The international audience may not understand the nuances or whether they can be compared with RNs.

THIS IS SO WRITTEN BECAUSE IN SOME JURISDICTIONS THE TERM ‘NURSE’ HAS VERY SPECIFIC MEANING AS A REGISTERED NURSE AND ONE HAS TO BE CAREFUL NOT TO OFFEND. WE HAVE AMENDED THE TERM TO MATCH THAT EXACTLY TO THE ONE USED IN THE PARTICIPANT SECTION OF THE PROTOCOL AND WHICH WE CLARIFIED IN OUR RESPONSE TO THE EARLIER REVIEWER COMMENT

Line 58. Is the Framework analysis reference incorporated into reference No. 52 cited here?

YES

P. 10, Line 40. Is there a risk of bias (selective reporting) if only some SAEs are reported and only if the CI deems them to be relevant?

THIS IS STANDARD OPERATING PROCEDURE FOR CLINICAL TRIALS, INCLUDING THOSE SUPPORTED BY CLINICAL TRIALS UNITS. THIS PROCESS IS OVERSEEN BY THE TRIAL SPONSOR AS WELL AS THE INDEPENDENT TRIAL STEERING COMMITTEE (TSC). WE HAVE ADDED A STATEMENT TO THE TEXT.

ALL SAEs, DEFINED IN THE SAME STANDARDISED MANNER HERE IN THIS TRIAL AS FOR ALL TRIALS UNDER GCP AND HRA REGULATIONS, WILL BE REPORTED BY SITES AND EXAMINED TO SEE IF THEY MEET THE CRITERIA TO BE CONSIDERED AS REPORTABLE, AND LIKELY TO BE RELATED TO TRIAL PROCEDURES OR NOT. BECAUSE, SADLY, SOME SERIOUS ADVERSE EVENTS UNRELATED TO THE TRIAL ARE EXPECTED (DETERIORATION AND DEATH) AS A CONSEQUENCE OF SOME PATIENT'S HEALTH CONDITION, THEY ARE NOT REPORTABLE TO THE TSC, UNLESS IT CAN BE ARGUED THAT THE TRIAL CONTRIBUTED TO THE EVENT. AS WE NOTE, THIS IS NOT A TRIAL OF A MEDICAL INTERVENTION AND UNLIKELY THAT THE INTERVENTION WILL LEAD TO SERIOUS ADVERSE EVENTS

P. 11, Line 8. Suggest to change to “responsive to participant needs.”
WE AMENDED THIS

Line 20. Suggest to change to “from participants who individually consent.”
WE AMENDED THIS

Lines 26-29. Repeated from earlier.
WE DELETED THE REPETITION REFERRED

Line 55. Should this be well developed NHS sites rather than through “experienced...”
NO, WE MEAN EXPERIENCED IN THIS CASE

P. 12.
Line 5. WHO COVID-19 R&D Roadmap needs a reference.
ADDED: (THIS WAS INCLUDED AT THE END OF THE SUBSEQUENT SENTENCE IN THE MANUSCRIPT BUT IS REPLICATED HERE FOR CLARITY)

Again, thank you for asking me to review this paper. In summary, I do have some concerns in terms of stated hypothesis, research question and/or aims. Without seeing the “COVID-specific clinical nursing care protocol” (p. 11, Line 43) referred to, I cannot evaluate whether this protocol is likely to be feasible or reproducible, as I would normally expect to when reading a protocol. With revision of the paper to include the above-mentioned components, the proposed protocol would likely be of interest to BMJ Open’s readers.

AS NOTED ABOVE, THIS IS A RESEARCH PROTOCOL, NOT A CLINICAL PROTOCOL, AND TO DIVULGE MORE CLINICAL DETAIL WOULD COMPROMISE THE TRIAL.

VERSION 2 – REVIEW

REVIEWER	Hosie, Annmarie The University of Notre Dame Australia, School of Nursing Sydney
REVIEW RETURNED	15-Mar-2021

GENERAL COMMENTS	Thank you for the opportunity to review your revised protocol, ‘COVID-NURSE: development and evaluation of the effects of a COVID-specific fundamental nursing care protocol compared to care as usual on experience of care for non-invasively ventilated patients in hospital with the SARS-CoV-2 virus: protocol for a cluster randomised controlled trial’. In brief, the revised manuscript is dramatically improved in terms of clarity, conciseness, and orderliness. It was almost effortless to read and comprehend. I also commend the authors on their detailed response to the reviewers’ comments and perseverance with the study despite identifying the need for significant design changes for sufficient and timely recruitment. With regards to the matter of transparently reporting the nursing
--

	protocol, I believe that it is scientifically and ethically necessary to do so, for these reasons:  1. It is customary to do so in the field of clinical research. 2. It is recommended and explained by the SPIRIT guidance: https://www.bmj.com/content/346/bmj.e7586.full?ijkey=QpAJnYI57zIwVr3&keytype=ref. Specifically, “Studies of trials and systematic reviews have shown that important elements of the interventions are not described in half of the publications. If such elements are also missing from the protocol, or if the protocol simply refers to other documents that are not freely accessible, then it can be impossible for healthcare providers, systematic reviewers, policymakers, and others to fully understand, implement, or evaluate the trial intervention. This principle applies to all types of interventions, but is particularly true for complex interventions (eg, health service delivery; psychotherapy), which consist of interconnected components that can vary between healthcare providers and settings.” 3. While the primary role of peer-reviewers of manuscripts is to comment on the reporting of a study, rather than its components or design, without knowing what the study intervention is, peer-review is hampered. 4. One of the primary reasons for protocol reporting – i.e. minimisation of reporting bias – is also threatening by absent a priori reporting of the intervention. 5. The numerous efforts being made in the intervention arm to support the implementation of the nursing protocol indicates that managers, site investigators and nurses in the control sites are unlikely to change practice or outcomes at their sites simply by reading the published clinical protocol. 6. Because the nursing care contained within the developed nursing protocol is fundamental care, ethically it belongs to all (even while your prior work to systematically review and articulate the components rightly is your intellectual property). 7. The components of fundamental care should already be known by nurses and nursing managers, and they are also already widely accessible online e.g. https://intlearningcollab.org/mission/the-fundamentals-of-care/ 8. By the time the protocol is published, it is likely most of the data collection will have occurred. 9. During the period of data collection, it is also likely that other related literature and evidence will be published and readily accessible. 10. Your point about blinding is an important one, but I believe the risk to blinding should simply be acknowledged, and can be justified by the need to maintain the other scientific and ethical imperatives outlined above. Other, more minor recommendations for the protocol report are:  1. Title: suggest a more concise and precise title is: ‘A COVID-19 fundamental nursing care protocol for non-invasively ventilated patients in hospital with the SARS-CoV-2 virus: protocol for a cluster randomised controlled trial’. 2. Regarding the varying use of COVID-19 and SARS-CoV-2 virus, throughout, is this deliberate? If so, why? As it seems clearer to just use COVID-19 throughout. On a related note, sometimes just ‘COVID’ is used, whereas the term ‘COVID-19’ is most correct. 3. The sentence starting ‘We also planned to...’ on page 8 is very long and hard to follow – suggest re-working this information into 2-3 sentences. 4. Regarding: “Control: care as usual only. We will record staffing
--	---

	staff skill mix details alongside a description of clinical nursing procedures and organisation in place in the usual care clusters.” – will you also document implementation initiatives independently undertaken at those sites? As it is likely they will occur. 5. Regarding: “Given this trial is not a clinical trial of an investigational medicinal product (CTIMP) the recording and reporting of non-serious adverse events (AEs) is not required (64)”, please explicitly state the research governance body permitting this to make it clearer that this decision comes from them and not from just the study team. 6. Please add a section explicitly stating the limitations of the study. 7. Please add a heading for the reference list. 8. While the report is very well written overall, a few typos and punctuation errors remain, including in the reference list – please check and correct these.
--	--

REVIEWER	Couper, Keith University of Warwick, Warwick Medical School
REVIEW RETURNED	12-Mar-2021

GENERAL COMMENTS	Thank you for responding to the first round of reviewer comments. I note the important changes in methodology and sample size. I am still not entirely as to what represents a cluster and the implementation of the intervention within that cluster. The cluster seems to be defined as the hospital trust, and recruitment will occur on COVID-19 wards within that trust. Where there is more than one COVID-19 ward in a trust, are hospitals required to implement the intervention across all COVID wards or can they select a single ward? I do have ongoing concerns as to your reticence to describe the intervention in more detail. I appreciate your keenness to avoid contamination. However, the value of a protocol publication is that it clearly sets out your trial design and intervention in a transparent way. I would encourage the authors to reflect on what additional information it might be reasonable to include. My comment on screening data linked to my concern of selection bias. To limit concerns about selection bias, it will presumably be important to understand how many patients in total were cared for on the ward and reasons why individuals were not approached (e.g. confusion). It may be helpful to clarify if you will be collecting these data. For some reason, the documents uploaded as appendices (as described in the response to reviewers) were not available to me. Were these uploaded?
---

REVIEWER	Kuhn, Lisa Monash University School of Nursing and Midwifery
REVIEW RETURNED	12-Apr-2021

GENERAL COMMENTS	Thank you for the opportunity to complete another review of this manuscript entitled “COVID-NURSE: development and evaluation of the effects of a COVID-specific fundamental nursing care protocol compared to care as usual on experience of care for non-invasively ventilated patients in hospital with the SARS-CoV-2 virus: protocol
--

	for a cluster randomised controlled trial.” The protocol for the proposed trial is registered, funded and Ethics’ approved. Overview Again, I believe that the paper is topical and will likely be of interest to BMJ Open’s readership. It has been adequately revised to make a positive contribution to this journal and patient care. The authors provide an extensive discussion about the planning and rationale for the comprehensive nursing-centred cluster randomised study regarding COVID-nursing care in non-invasively ventilated hospital patients in the United Kingdom. There are no typographical errors obvious throughout the paper and the referencing is consistent and contemporary. Authors have now been listed in the PROSPERO study reference. I have carefully read my previous hand-written notes on the first version I reviewed, along with all reviewer comments and the author responses available to me. The paper is now clear to read and the additional explanations regarding areas such as strengths and limitations, rationale for amending the method and experience so far, sample size, timelines, differences between registered nursing and other care staff, missing data and evaluations are excellent. The objectives are prominent and the explanation for focusing on black and Asian ethnicity people in particular is well explained and justified. The Gantt chart and flow chart provided are useful. Very minor points include:  • BAME needs to be spelled out at first use in the Methods section • The first few sentences under the heading Safety reports of serious adverse events (SAE) need tweaking grammatically. See “is not required(64) since, although...”, and “but that these will be both...” • There’s an ‘&’ in between Hayes and Moulton (16) under the Target sample size subheading.
--	--

VERSION 2 – AUTHOR RESPONSE

Reviewer: 1
Dr. Annmarie Hosie, The University of Notre Dame Australia

Comments to the Author:

Thank you for the opportunity to review your revised protocol, ‘COVID-NURSE: development and evaluation of the effects of a COVID-specific fundamental nursing care protocol compared to care as usual on experience of care for non-invasively ventilated patients in hospital with the SARS-CoV-2 virus: protocol for a cluster randomised controlled trial’.

In brief, the revised manuscript is dramatically improved in terms of clarity, conciseness, and orderliness. It was almost effortless to read and comprehend. I also commend the authors on their detailed response to the reviewers’ comments and perseverance with the study despite identifying the need for significant design changes for sufficient and timely recruitment.

With regards to the matter of transparently reporting the nursing protocol, I believe that it is scientifically and ethically necessary to do so, for these reasons:

1. It is customary to do so in the field of clinical research.
2. It is recommended and explained by the SPIRIT guidance:
<https://www.bmj.com/content/346/bmj.e7586.full?ijkey=QpAJnYI57zIwVr3&keytype=ref>. Specifically,

“Studies of trials and systematic reviews have shown that important elements of the interventions are not described in half of the publications. If such elements are also missing from the protocol, or if the protocol simply refers to other documents that are not freely accessible, then it can be impossible for healthcare providers, systematic reviewers, policymakers, and others to fully understand, implement, or evaluate the trial intervention. This principle applies to all types of interventions, but is particularly true for complex interventions (eg, health service delivery; psychotherapy), which consist of interconnected components that can vary between healthcare providers and settings.”

3. While the primary role of peer-reviewers of manuscripts is to comment on the reporting of a study, rather than its components or design, without knowing what the study intervention is, peer-review is hampered.

4. One of the primary reasons for protocol reporting – i.e. minimisation of reporting bias – is also threatening by absent a priori reporting of the intervention.

5. The numerous efforts being made in the intervention arm to support the implementation of the nursing protocol indicates that managers, site investigators and nurses in the control sites are unlikely to change practice or outcomes at their sites simply by reading the published clinical protocol.

6. Because the nursing care contained within the developed nursing protocol is fundamental care, ethically it belongs to all (even while your prior work to systematically review and articulate the components rightly is your intellectual property).

7. The components of fundamental care should already be known by nurses and nursing managers, and they are also already widely accessible online e.g. <https://intlearningcollab.org/mission/the-fundamentals-of-care/>

8. By the time the protocol is published, it is likely most of the data collection will have occurred.

9. During the period of data collection, it is also likely that other related literature and evidence will be published and readily accessible.

10. Your point about blinding is an important one, but I believe the risk to blinding should simply be acknowledged, and can be justified by the need to maintain the other scientific and ethical imperatives outlined above.

We have addressed this substantive issue in our response to the editorial requirements above and have greatly expanded our description of the clinical intervention as a consequence. We hope this will be sufficient to address the reviewer’s request to provide more description of our clinical intervention.

Additionally, our response to the reviewer would be that we do consider some of the points being made as overstated and some incorrect. We detail our response to these specifically below.

In response to point 2) we are fully aware and support open access intervention transparency efforts such as TiDiER and will most certainly make our full clinical guideline and educational resource available post-trial in all its complete detail. We have now made this clear in the manuscript. We have never suggested otherwise. The online course will be available as a free to access MOOC. Our position has never been one of disputing this principle, far from it, but nervousness about ethical trial conduct. Once complete, all materials will be available.

In point 5), this is an empirical question. We disagree with the reviewer in that we cannot assume, as the reviewer suggests, that supportive strategies are effective. We must first analyse process data on implementation mechanisms. We are collecting such data in our trial. It is perfectly possible that our supportive implementation efforts are redundant and that nurses will be hungry for a guideline *per se*, given we know that no evidence-based guidance exists elsewhere. Our process evaluation will answer this question.

In point 6) the reviewer is suggesting we are claiming intellectual property rights over fundamental care. We are disappointed that this suggestion has been made and cannot let it stand unchallenged. This is a publicly funded study whose results and materials we will make available free of cost globally as well as nationally in the UK – again we reiterate that this is very clear now in the manuscript. However, at no point have we ever suggested otherwise. Our reasons for reticence in publicising the clinical materials at this point have only, and solely, been in order to preserve trial integrity, not to claim intellectual property rights, or to prevent widespread dissemination to, and use of our work by, hard pressed

clinical services. Indeed, some of our investigator team deliver and manage services themselves.

In point 7, the reviewer is suggesting that since the components of fundamental nursing care are available elsewhere, there is redundancy in us testing them. In fact, the chief investigator and corresponding author of this COVID NURSE study protocol is the UK representative on the board of the organisation (the ILC) cited by the reviewer above. We are, therefore, well aware of the materials referred to by the reviewer. It is precisely because these exist only at a theoretical level that trials such as ours are vital to turn these theories into practical tools for nurses. We are sadly certain that only a very small minority of practising clinical nurses are even aware of this organisation and its work. Implementation Science has taught us time and time again that information alone, particularly online, is insufficient to change clinical practice. We, therefore, disagree with the reviewer's point.

Point 8 we have dealt with above. Point 9 is highly conjectural (we are not aware of any other trials of fundamental nursing care for patients admitted to hospital with a SARS-COV-2 infection). Even if other studies emerge, we do not see the logic of the point being made.

Finally, we strongly contest the suggestion from the reviewer that we are behaving unethically in preserving the integrity of our intervention blinding. As noted earlier, we are trying to balance two conflicting ethical principles. Compromised blinding is a criterion for rating a trial as subject to potential bias – itself an ethical issue when we are asking participants to engage voluntarily in RCTs. Trials with compromised blinding are often rejected in systematic reviews or lead to uncertain conclusions. We consider it an ethical principle we act to ensure all participants in our trial contribute their data to a study that is neither compromised nor subject to significant potential biases.

Other, more minor recommendations for the protocol report are:

1. Title: suggest a more concise and precise title is: 'A COVID-19 fundamental nursing care protocol for non-invasively ventilated patients in hospital with the SARS-CoV-2 virus: protocol for a cluster randomised controlled trial'.

We have amended the title slightly to reduce wordage as suggested, albeit we have retained the original PICO structure

2. Regarding the varying use of COVID-19 and SARS-CoV-2 virus, throughout, is this deliberate? If so, why? As it seems clearer to just use COVID-19 throughout. On a related note, sometimes just 'COVID' is used, whereas the term 'COVID-19' is most correct.

Yes, this is deliberate. We use SARS-COV-2 to describe the viral agent and COVID-19 the syndrome it causes, which we believe to be the correct usage. We only used COVID without '19' twice in the terms 'COVID-specific'. We have added 19 to these two instances, albeit it does look a little clumsy.

3. The sentence starting 'We also planned to...' on page 8 is very long and hard to follow – suggest re-working this information into 2-3 sentences.

Thank you, we have edited this section to be clearer.

4. Regarding: "Control: care as usual only. We will record staffing staff skill mix details alongside a description of clinical nursing procedures and organisation in place in the usual care clusters." – will you also document implementation initiatives independently undertaken at those sites? As it is likely they will occur.

This will be extremely difficult to achieve, although we will certainly seek to describe any such amendments to clinical care in control sites.

5. Regarding: "Given this trial is not a clinical trial of an investigational medicinal product (CTIMP) the recording and reporting of non-serious adverse events (AEs) is not required (64)", please explicitly

state the research governance body permitting this to make it clearer that this decision comes from them and not from just the study team.

As noted in the manuscript, we have obtained ethical approval from the UK Health Research Authority Research Ethics Committee system, which includes as a matter of course, analysis and approval of our SAE procedures. We have added the details of the HRA REC committee to the text as requested.

6. Please add a section explicitly stating the limitations of the study.

We have added this, although it replicates the bullet pointed section required by the BMJ Open guidance. We will leave it as an editorial decision whether this replication is required or not.

7. Please add a heading for the reference list.

We have added this

8. While the report is very well written overall, a few typos and punctuation errors remain, including in the reference list – please check and correct these.

We have spotted some of these and edited them in the text and in the reference manager system we use. Hopefully, there are no others but we will of course respond to any queries raised during proofing.

Reviewer: 2

Dr. Keith Couper, University of Warwick

Comments to the Author:

Thank you for responding to the first round of reviewer comments. I note the important changes in methodology and sample size.

I am still not entirely as to what represents a cluster and the implementation of the intervention within that cluster. The cluster seems to be defined as the hospital trust, and recruitment will occur on COVID-19 wards within that trust. Where there is more than one COVID-19 ward in a trust, are hospitals required to implement the intervention across all COVID wards or can they select a single ward?

We have clarified this point under the section 'Setting'

I do have ongoing concerns as to your reticence to describe the intervention in more detail. I appreciate your keenness to avoid contamination. However, the value of a protocol publication is that it clearly sets out your trial design and intervention in a transparent way. I would encourage the authors to reflect on what additional information it might be reasonable to include.

We have addressed this above and hope that we have provided enough detail now

My comment on screening data linked to my concern of selection bias. To limit concerns about selection bias, it will presumably be important to understand how many patients in total were cared for on the ward and reasons why individuals were not approached (e.g. confusion). It may be helpful to clarify if you will be collecting these data.

As part of the trial procedures we have a recruitment log which details potential participants approached. This data will assist in addressing this point. However, it will be extremely difficult to exactly determine the sample denominator. We will seek to gain as accurate an estimate of this as possible from occupancy rates in sites during the data collection period per site.

For some reason, the documents uploaded as appendices (as described in the response to reviewers) were not available to me. Were these uploaded?

We apologise and have now remedied the mistake

Reviewer: 3

Dr. Lisa Kuhn, Monash University School of Nursing and Midwifery

Comments to the Author:

Dear Author team,

Thank you for the opportunity to complete another review of this manuscript entitled “COVID-NURSE: development and evaluation of the effects of a COVID-specific fundamental nursing care protocol compared to care as usual on experience of care for non-invasively ventilated patients in hospital with the SARS-CoV-2 virus: protocol for a cluster randomised controlled trial.”

The protocol for the proposed trial is registered, funded and Ethics’ approved.

Overview

Again, I believe that the paper is topical and will likely be of interest to BMJ Open’s readership. It has been adequately revised to make a positive contribution to this journal and patient care. The authors provide an extensive discussion about the planning and rationale for the comprehensive nursing-centred cluster randomised study regarding COVID-nursing care in non-invasively ventilated hospital patients in the United Kingdom.

There are no typographical errors obvious throughout the paper and the referencing is consistent and contemporary. Authors have now been listed in the PROSPERO study reference.

I have carefully read my previous hand-written notes on the first version I reviewed, along with all reviewer comments and the author responses available to me. The paper is now clear to read and the additional explanations regarding areas such as strengths and limitations, rationale for amending the method and experience so far, sample size, timelines, differences between registered nursing and other care staff, missing data and evaluations are excellent. The objectives are prominent and the explanation for focusing on black and Asian ethnicity people in particular is well explained and justified. The Gantt chart and flow chart provided are useful.

Very minor points include:

- BAME needs to be spelled out at first use in the Methods section

Thank you, we have now done so

- The first few sentences under the heading Safety reports of serious adverse events (SAE) need tweaking grammatically. See “is not required(64) since, although...”, and “but that these will be both...”

We have re-read this and in fact we believe that the grammar is actually correct. If not, we would be happy to amend at the sub-editing/proofing stage.

- There’s an ‘&’ in between Hayes and Moulton (16) under the Target sample size subheading.

This has been removed and replaced with ‘and’.